# Review of Evidence Available on Hesperidin-Rich Products as Potential Tools against COVID-19 and Hydrodynamic Cavitation-Based Extraction as a Method of Increasing Their Production

**Francesco Meneguzzo** [1,*] **, Rosaria Ciriminna** [2] **, Federica Zabini** [1] **and Mario Pagliaro** [2]

[1] Institute for Bioeconomy, National Research Council, 10 Via Madonna del Piano, I-50019 Sesto Fiorentino, Italy; federica.zabini@cnr.it

[2] Institute for the Study of Nanostructured Materials, CNR, via U. La Malfa 153, I-90146 Palermo, Italy; rosaria.ciriminna@cnr.it (R.C.); mario.pagliaro@cnr.it (M.P.)

* Correspondence: francesco.meneguzzo@cnr.it; Tel.: +39-392-985-0002

**Abstract:** Based on recent computational and experimental studies, hesperidin, a bioactive flavonoid abundant in citrus peel, stands out for its high binding affinity to the main cellular receptors of SARS-CoV-2, outperforming drugs already recommended for clinical trials. Thus, it is very promising for prophylaxis and treatment of COVID-19, along with other coexistent flavonoids such as naringin, which could help restraining the proinflammatory overreaction of the immune system. Controlled hydrodynamic cavitation processes showed the highest speed, effectiveness and efficiency in the integral and green aqueous extraction of flavonoids, essential oils and pectin from citrus peel waste. After freeze-drying, the extracted pectin showed high quality and excellent antioxidant and antibacterial activities, attributed to flavonoids and essential oils adsorbed and concentrated on its surface. This study reviews the recent evidence about hesperidin as a promising molecule, and proposes a feasible and affordable process based on hydrodynamic cavitation for the integral aqueous extraction of citrus peel waste resulting in hesperidin-rich products, either aqueous extracts or pectin tablets. The uptake of this process on a relevant scale is urged, in order to achieve large-scale production and distribution of hesperidin-rich products. Meanwhile, experimental and clinical studies could determine the effective doses either for therapeutic and preventive purposes.

**Keywords:** citrus fruits; coronavirus; COVID-19; flavonoids; hesperetin; hesperidin; hydrodynamic cavitation; pectin; SARS-CoV-2

## 1. Introduction

The pandemic caused by the spreading of the severe acute respiratory syndrome coronavirus 2 (SARS-CoV-2) and the related disease named coronavirus disease 2019 (COVID-19), from China since late 2019 to most of the world since January 2020, proved unprecedented in many aspects. A combination of the high contagiousness rate (even from asymptomatically infected people), the long course of the disease, relatively high incidence of severe and lethal pneumonia (especially for elderly and immunosuppressed subjects) [1], lack of effective therapies, and initial unpreparedness to cope with the pandemic threaten to overwhelm local and national health care infrastructures—intensive care units in particular [2,3]. The consequent reaction concretized literally overnight, expanding to a remarkable fraction of the world population in total or partial lockdowns and social distancing prescriptions, which are still in force at the time of writing. While everyday chronicles proved the effectiveness of

such measures in slowing down the infection rate, they are also likely to cause unpredictable economic havoc and endanger the livelihood of many people for a long time [4].

While similar to other coronaviruses responsible for past epidemics, especially SARS-CoV (early 2000s), with respect to targeted receptors of human type II lung cells [5,6], genetic mutations that occurred in SARS-CoV-2 made it more infectious and likely more effective in slowing the response by the immune system until the virus has spread in the lung cells and has started replicating [7]. In particular, long spike glycoproteins that protrude from the SARS-CoV-2 particle latch on to the angiotensin converting enzyme-2 (ACE2), a protein located on the surface of type II lung cells [6].

As with SARS-CoV, most of the damage in COVID-19 is caused by the immune system carrying out an overreaction to stop the virus from spreading. Upon entry into alveolar epithelial cells, SARS-CoV-2 replicates rapidly and triggers a strong immune response, resulting in cytokine storm syndromes, or hypercytokinemia, and pulmonary tissue damage. The uncontrolled production of proinflammatory cytokines (and chemokines) causes acute respiratory distress and multiple organ failure [8,9], even possibly affecting the male gonadal function [10]. Beyond the lethal cases, it is still unclear whether and at which extent these damages could be reversed in recovered subjects.

An unprecedented worldwide scramble is underway to search for effective vaccines [11]. However, their very feasibility and effectiveness is still uncertain. Recent preliminary results point to insufficient development of SARS-CoV-2-specific neutralizing antibodies in a fraction of recovered patients, especially the younger or those affected by common or mild symptoms [12]. The same finding, if confirmed, might reduce the expectancies about the perspective for herd immunity.

An intensive research also is underway to identify therapeutic drugs to be repurposed, a few of which could already have shown preliminary positive results, yet lack large and randomized verifications and consideration of possible harmful side effects [13]. Natural bioactive compounds are also actively looked for, to assess their preventive or therapeutic activities, namely the ability to prevent the virus from binding to the ACE2 enzyme of the host cell, inhibit the virus replication after its penetration in the host cell, as well as restrain or counteract the proinflammatory overreaction of the immune system.

The frantic search for effective bioactive compounds is part of the general vision that perceives the boost to the individual immune system, and the effective inhibition of the infection by SARS-CoV-2 as the most effective shield against the onset and serious progress of COVID-19, protecting both individuals and society as a whole [14,15]. Contributing to transform this vision into reality is also the goal of this study.

Past and recent studies proved that hesperidin, a citrus flavonoid abundant in citrus peel, which is a byproduct of the juice industry, as well as the major flavonoid in sweet orange and lemon, is endowed with plenty of beneficial biological activities, some of which are shared with other citrus flavonoids. Thus, it is not surprising that many food supplements and drugs containing hesperidin and other citrus flavonoids have been available since long. Hesperidin and its aglycone hesperetin were attributed particularly strong binding affinity to the receptors of SARS-CoV-2, along with remarkable anti-inflammatory activity, making these molecules attractive ingredients for preventive and therapeutic drugs.

In Section 2, the sources and remarkable bioactive properties, including antiviral activity, of hesperidin and other citrus flavonoids are briefly reviewed, along with the challenging issue posed by generally low bioavailability. Section 3 focuses on the properties of hesperidin and other citrus flavonoids potentially relevant to the contrast to COVID-19. Section 4 reviews the extraction methods of the same compounds, pointing to controlled hydrodynamic cavitation (HC) as the most effective, efficient and scalable in the perspective of large-scale production. The discussion and conclusions in Section 5 highlights the immediate feasibility of mass production of hesperidin-rich products based on existing plants and their upscale and replication, also suggesting an affordable process line.

## 2. Sources and Bioactive Properties of Hesperidin and Other Citrus Flavonoids

### 2.1. Sources of Hesperidin

Hesperidin (hesperetin 7-rutinoside) is a flavanone glycoside composed of hesperetin (an aglycone unit) and rutinose (a disaccharide). First isolated in 1828, hesperidin is the dominant flavonoid in citrus species, occurring mainly in the peel of oranges (*Citrus sinensis*) and lemons (*Citrus limon*) [16].

The content of hesperidin in citrus fruits shows remarkable dependence on the variety of fruit [17,18]. For example, sweet oranges juices contain about 200–600 mg/L of hesperidin, clementines 50–850 mg/L, mandarins 8.1–460 mg/L, lime and lemon juice 38–410 mg/L, and 20–170 mg/L in grapefruit juice [19].

The concentration of hesperidin changes as well across the different parts of a fruit, with far higher levels in citrus peel (flavedo, albedo, membrane and pith) than in the other parts of the citrus fruits (juice vesicles and seeds). The observed concentration in the albedo (about 15 mg/g fresh weight) was as much as 16 times higher than in juice vesicles, with similar ratios holding for other citrus flavonoids, such as naringin [20]. Observed reference levels for the concentration of hesperidin in orange peels were 2.1 mg/g (fresh weight) and 48 mg/g (dry weight) [21]. The concentration of hesperidin is also related to the growth stages of the fruit [22], as well to the specific growing sites [23,24].

Besides fruits, hesperidin is available in other parts of the citrus plant, such as leaves, with higher and remarkable concentration in young leaves (about 50% the concentration found in the fruit's albedo) [20]. Hesperidin occurs in notable concentrations in several plants other than citrus, including certain officinal plants, such as *Menthae piperitae*, *Hypericum perforatum* and *Salvia officinalis*, although less abundant than other flavonoids such as rutin [25]. In the case of *Menthae piperitae*, a wide range of levels was observed, depending on the varieties and growing sites, such as up to 2.15 mg/g (dry weight) [26], and up to 13.1 mg/g (dry weight) [27]. Moreover, the remarkable level of 26.9 mg/g (dry weight) was found in the stems of *Cyclopia maculate*, which are byproducts from herbal tea processing [21]. Hesperidin, along with neohesperidin, was detected in many other plants, and parts thereof, growing in different areas of the world [16].

### 2.2. Safety and Broad Spectrum Activities

Empirically known since 1876, when the beverage hesperidin obtained from bitter and sweet orange peels was first introduced in Argentina [28], the numerous and different health beneficial effects of hesperidin allowed the exploitation of its large pharmaceutical potential [29].

Human studies have shown since long that the substance is safe and well tolerated up to very high doses of administration. For example, in 1964 a study during which 94 menopausal women had a daily intake of 900 mg of hesperidin (in addition to 300 mg of hesperidin methyl chalcone and to 1200 mg of vitamin C) for 1 month demonstrated the safety of hesperidin, even at such high dosage [30]. In murine studies, orally administered doses of up to 5% (5 g per 100 g of body weight) showed no toxicity [16].

Among commercial products, the flavonoid vasoprotector and venotonic agent Daflon 500 mg, which has been commercialized for more than 25 years, contains 450 mg of diosmin (90%) and 50 mg of hesperidin (10%), and has been proved effective and safe in the long term [31]. Many other products, mainly food supplements, containing hesperidin (up to 500 mg per tablet) and other citrus flavonoids are available on the market and regularly consumed since long, generally claiming to contrast the capillary fragility, and producing antiedema and anti-inflammatory activity.

The broad spectrum of biological activities of hesperidin and other citrus bioflavonoids have been known since the 1930s and were also comprehensively reviewed two decades ago based on commercial products [16]. Hesperidin was attributed strong antioxidant activity, significant effects on the vascular system, in particular decreasing the capillary permeability and increasing the capillary resistance. Associated with naringin (another citrus flavonoid), hesperidin was shown to significantly lower levels of plasma, hepatic cholesterol, and hepatic triglycerides. Through various mechanisms, hesperidin and other flavanones were shown effective against hypertension. Associated with diosmin—another

flavonoid glycoside, and through various mechanisms, hesperidin showed marked protective effect against inflammatory disorders. For example, the agent Daflon 500 mg (diosmin 90%, hesperidin 10%) was demonstrated to improve multiple histological aspects of the acute inflammatory reaction as well as of the chronic inflammation. The sulphonated and phosphorylated hesperidin compounds proved to be extremely potent inhibitors of the hyaluronidase enzyme, which causes a breakdown of hyaluronic acid, thereby increasing tissue permeability and favoring the penetration of certain harmful bacteria.

In a 2011 large clinical study, hesperidin displayed a relevant role in the genomic effect of orange juice, resulting in significant anti-inflammatory and antiatherogenic activities [32]. A daily dose of 292 mg of hesperidin, corresponding to 500 mL of orange, was sufficient to display the aforementioned effects.

In a recent comprehensive review, all the above-mentioned effects were confirmed and updated [29]. A result was particularly relevant to this study: hesperidin and its aglycone hesperetin, the latter being relatively scarce in citrus fruits and also derived from hesperidin by means of intestinal bacteria following ingestion [16], were found effective to dwindle the release of proinflammatory cytokines from immune cells in several tissues, including cerebral, kidney, blood, and lungs. Hesperidin showed to be an effective antagonist of Th2 cytokine in the alveolar space, where localized inflammatory cytokine storms occur in the early phase of acute respiratory distress syndrome and are associated with profibrotic collagen synthesis, sometimes leading to the permanent replacement of the original tissue with scar tissue and eventually organ damage or failure [33,34].

In a study on cancer-induced cachexia (unintentional loss of body weight and skeletal muscle), an integral water extract of *Citrus unshiu* peel showed effective in restraining the cachexia effects by means of the efficient suppression of the production of procachectic cytokines in immune cells as well as cancer cells [35]. Hesperidin revealed the most effective molecule out of all the citrus flavonoids. On the other hand, hesperidin was studied as a potentially effective cancer chemoprotective agent, mainly due to is potent antioxidant activity and scavenging of reactive oxygen species, although still limited clinical trials prevent clear conclusions from being drawn [36].

Also relevant to this study, hesperidin was shown particularly effective against retinopathy induced by oxidative stress. This effect was attributed to its antioxidant activity and the suppression of excessive activation of calpain, a cysteine protease [37]. Hesperidin was also found very effective in the protection from diabetic retinopathy [38].

### 2.3. Antiviral Activity

Specific antiviral activity of hesperidin and its aglycone hesperetin has long been known, based on in vitro studies, especially towards influenza virus and some herpes viruses [16]. Hesperetin was attributed inhibition activity against the replication of the same herpes viruses. Hesperidin showed also a potent inhibitory effect on the infectivity of rotavirus, both isolated and in integral extracts.

In murine experiments, hesperidin, at a dose of 100 mg per kg of body weight and with intragastric administration, was found to effectively inhibit influenza A virus replication and spread, by up-regulating certain cell-autonomous immune responses [39]. Conversely, other flavonoids, such as kaempferol, induced down-regulation and promoted virus replication. Following previous studies that related the variety of the physiological activities of flavonoids to their structure and geometry [40], the opposite effects of hesperidin and kaempferol on influenza virus replication were tentatively attributed to structural differences [39]. In particular, flavonoids possessing an ability of anti-influenza virus replication have a double hydrogen bond between C2 and C3 and a replaced group at C2, or a double bond between C2 and C3 and a replaced group at C3. In contrast, the flavonoids promoting influenza virus replication all have a double bond between C2 and C3 and a replaced group at C2, suggesting that the form of chemical bond between C2 and C3 and a replaced group at C2 or C3 in one flavonoid compound may be essential for its anti-influenza virus efficacy.

The flavonoid glycosides hesperidin and linarin, the latter derived from certain herbs, share common features such as rutinose at the A ring and methoxy (-OCH3) substitution at the B ring. A recent study showed that hesperidin and linarin were very effective, in a dose-dependent manner, in suppressing the replication of the R5-type of human immunodeficiency virus (HIV-1), which remain in the ileum of patients even after treatment with the most effective antiretroviral drugs [41]. The mechanism of action was identified in the stimulation of peripheral blood mononuclear cells and the consequent secretion of certain cytokines. A notable result was that the rutinose-deficient analogs of hesperidin (its aglycone hesperetin) and linarin (acacetin), as well as other flavonoids lacking a methoxy substitution at the B ring, did not show similar effects, or only very attenuated ones.

In the aftermath of the SARS-CoV epidemic of the early 2000s, as early as 2005, hesperetin was found to be the most effective molecule, out of synthetic and other natural products (hesperidin was not included), in the inhibition of the SARS-CoV 3-chymotrypsin-like protease (3CL$^{pro}$). It showed a level of the 50% inhibitory concentration (IC$_{50}$) of 8.3 μM in the cell-based assay, much smaller than the next natural molecule aloe emodin [42,43]. The 3CL$^{pro}$, as a virus-encoded protease, mediates the proteolytic processing of certain replicase polypeptides into functional proteins, thus allowing the virus replication in the host cells and becoming an important target for the drug development. Chloroquine, a long known antimalarial drug, showed a slightly higher level of IC$_{50}$ of 8.8 μM [44]. Moreover, hesperetin turned out to be the most selective among the other considered molecules, thus showing the lowest level of cytotoxicity. It showed a remarkably high selectivity index (the ratio of the concentration of the compound that reduced cell viability to 50%, or CC$_{50}$, to the concentration needed to inhibit the viral cytopathic effect to 50% of the control value), of about 300, which was tenfold the level for chloroquine (CC$_{50}$ = 30).

The spike protein of SARS-CoV was identified as a general target for vaccines and therapeutic treatments [45]. A subunit (S1) of the spike protein contains a receptor-binding domain (RBD) that engages with the host cell receptor ACE2, while the other subunit (S2) mediates fusion between the viral and host cell membranes. However, the search for compounds effectively blocking the RBD–ACE2 binding and the spike protein-mediated infection, and/or the fusion of membranes of the virus and the host cell did not lead to conclusive results for SARS-CoV.

## 2.4. The Bioavailability Issue

The crystal structure of hesperidin, also common to naringin and the respective aglycones, hesperetin and naringenin, was deemed responsible for poor water solubility, along with the hydrophobic nature of the molecule itself and the primary particle size (generally greater than 5 μm) [46]. The remarkable size of primary particles is due, in turn, to the tendency of hesperidin to form complex crystals with other similar glucosides, which makes also difficult to obtain it in a pure state [16]. The solubility of hesperidin in water, pure or aggregated with other glucosides, is lower than 20 mg/L [47].

The water solubility issue was shown to limit the bioavailability of hesperidin, as well as its metabolic stability and spreading to tissues and organs, in particular after oral administration [16]. The low bioavailability level (<25%) of hesperidin was attributed to its irregular absorption in the gastrointestinal tract, as well as to the hydrolysis into aglycone hesperetin under gastric pH conditions, and enzymatic degradation [48], leading to striking differences among the physiological effects revealed in vitro and in vivo. The improvement of the water solubility of hesperidin was the subject of several studies, which aimed at creating more effective formulations for the delivery of hesperidin to physiological targets.

The most common method to enhance the bioavailability of hesperidin, as well as of other poorly available drugs, was to enhance their dissolution rate by coupling the bioactive compound with a suitable carrier able to destroy the crystalline structure, and forming solid fine dispersions, associated with larger available surface and more rapid wetting and dissolution [46]. This way, the drug can be released efficiently as very fine colloidal particles with a size less than 1 μm.

Another method consists in processing bioactive molecules in order to obtain more soluble molecules, while retaining the original properties. This is the case of glucosyl hesperidin, obtained by regioselective transglycosylation of hesperidin with cyclodextrin glucanotransferase from *Bacillus stearothermophilus*, which showed ten thousand times higher water solubility (around 200 g/L) and about 3.7 higher bioavailability, estimated through the concentration of its metabolite hesperetin in the sera of rats, while retaining the biological properties of hesperidin [49].

Hesperidin gastroresistant microparticles for oral administration were produced by spray-drying using cellulose acetate phthalate as a polymeric carrier with adequate coating properties, able to protect hesperidin in the gastric medium [48]. Avoiding gastric degradation, and increasing the solubility, this method allowed a very effective delivering of hesperidin to the intestine, thus improving its bioavailability when administered in solid oral form.

More recently, complexing of hesperidin with biological or mineral compounds was proposed as a means to increase solubility and bioavailability. Complexing hesperidin with natural products extracted from black tea, through pi–pi interactions and/or hydrophobic effects, led to an increase in solubility by up to 2.5-fold [50]. In another study, a 6-h continuous stirring at 2400 rpm and 50 °C of a mixture of pure water, hesperidin and modified sweet potato starch, with unadjusted pH, allowed hesperidin to penetrate into the modified starch cavity, resulting in a 6.52-fold increase of the solubility of hesperidin [51].

In a hydroalcoholic extraction of hesperidin from lime peel, the addition of 10% dimethyl sulfoxide (DMSO) allowed for a substantial increase in the solubility of hesperidin and its adsorption into resins [47]. As the DMSO is one of the least toxic organic solvents, with low chronic and acute oral toxicity, the authors proposed its use to deliver low solubility drugs, including hesperidin.

Mixing amide pectin with hesperidin in a water solution, then with chitosan in a hydroalcoholic solution, hesperidin was effectively entrapped inside the resulting hydrogel matrix [52]. In vitro experiments showed the remarkable mucoadhesive properties of the hydrogel to rat *caecum*, along with the ability to regulate the release of hesperidin to the target cells, up to 56% of the active substance.

Polyphenol-conjugated pectins (catechin, quercetin, rutin, and hesperidin as polyphenols) were synthesized as a result of epichlorohydrin conjugation reactions between polyphenol molecules and pectin in aqueous conditions [53,54]. The conjugates demonstrated remarkably higher solubility in water in comparison with neat pectin and polyphenols, as well as retained the original biological properties of the conjugated polyphenol.

Recently, a golden nanoparticle–conjugated hesperidin complex was produced and evaluated with respect to the stimulation of the activity of phagocytic cells [55]. The phagocytic index of the conjugated complex against *Staphylococcus aureus* bacteria was considerably higher (85%) in comparison with both gold nanoparticles (66%), hesperidin (51%) and untreated blood sample used as the control (20%), demonstrating its higher bioavailability.

In spite of its limited bioavailability, a recent in vivo murine study showed for the first time that hesperidin, after a single oral ingestion at the moderate dose of 10 mg/kg, could cross the intestinal membrane through the Caco-2 cell monolayers in its intact form and enter into the portal vein blood [56]. The concentration of hesperidin peaked 2 h after ingestion, at the level of about 0.06 nmol/mL-plasma. Subsequently, hesperidin was metabolized into its aglycone hesperetin, peaking in the blood system 8 h after ingestion.

In light of the potential applications of certain citrus flavonoids against pulmonary diseases, inhalable formulations were designed with the aim of directly targeting lung cells, while limiting the gastrointestinal degradation processes. While information on hesperidin is scarce, naringin was successfully formulated as an inhalable dry powder by spray-drying with the amino acids leucine, histidine and proline, resulting in a substantial reduction of the size distribution of particles and increased in vitro anti-inflammatory activity [57]. In particular, more than 60% of the spray-dried naringin particles achieved aerodynamic diameters <5 μm, which is considered representative of the

amount of the bioactive compound capable to reach the lung region in vivo, making this formulation suitable for inhalation therapy.

## 3. Early Evidence of Potential Activity against SARS-CoV-2

According to recent studies, hesperidin showed remarkable binding affinity to the three main protein receptors of SARS-CoV-2, i.e., the SARS-CoV-2 protease domain, the receptor binding domain of the spike glycoprotein (RBD-S), and the receptor binding domain of the ACE2 at the protease domain (RBD-ACE2), responsible for cell infection and virus replication [58–63]. The above-mentioned remarkable binding affinity to the three main targets was considered representative of the inhibitory activities of hesperidin against viral infection, by either inhibiting the latching of the virus to the ACE2, or inhibiting the virus replication in the cells. Thus, hesperidin could be a promising active substance for drugs potentially useful to prevent or treat COVID-19, possibly along with other citrus flavonoids.

In a molecular docking study, scholars in Indonesia found that hesperidin had the highest affinity to bind all three receptors (lowest docking score), thus inhibiting the proteins responsible for viral infection and virus development [58]. Hesperidin outperformed lopinavir, a repurposing drug involved in clinical trials for COVID-19, as well as nafamostat, a reference compound for RBD-S binding. Moreover, hesperidin outperformed several other natural molecules. In the same study, other citrus flavonoids also abundant in citrus peel, namely tangeretin, nobiletin and naringenin, as well as hesperetin that derives from hesperidin in the intestine after ingestion, showed excellent affinity to the selected receptors, suggesting that all these citrus flavonoids might contribute to inhibit the viral infection and replication.

Hesperetin was the only citrus flavonoid among the flavonoids investigated in an another study [64]. It showed high binding affinity to ACE2 enzyme, similar to other flavonoids typical of Chinese medicine, present in various herbs, roots, and soybean.

In another study, scholars in China reached similar conclusions [59]. In detail, the team analyzed all the proteins encoded by SARS-CoV-2 genes, compared them with other coronaviruses, such as SARS-CoV and MERS-CoV, and modeled the protein structures using said structures along with those of human relative proteins (human ACE2 and type-II transmembrane serine protease enzymes) as targets to screen three databases of approved drugs. These databases were the following: the database of traditional Chinese medicine and natural products (including reported common antiviral components from traditional Chinese medicine), the database of commonly used antiviral drugs (78 compounds), and the ZINC drug database of the Food and Drug Administration of the USA by virtual ligand screening method. The method clearly showed that hesperidin was the only compound that could target the binding interface between the spike protein and human ACE2, so that by superimposing the RBD–ACE2 complex to the hesperidin–RBD complex, a distinct overlap of hesperidin with the interface of ACE2 was observed. This suggests that hesperidin may disrupt the interaction of ACE2 with RBD and prevent the virus from entering the cell.

In a further study, a molecular model was built of the 3-chymotrypsin-like protease ($M^{pro}$/$3CL^{pro}$) structure of the SARS-CoV-2, which is vital to virus replication (as it was for SARS-CoV) and is considered as a promising drug target [60]. The study carried out virtual screening to identify readily usable therapeutics derived from the previous progress about specific inhibitors for the corresponding SARS-CoV enzyme [42–44], which can be conferred on its SARS-CoV-2 counterpart. Results showed that the flavonoid glycosides diosmin (a preapproved drug) and hesperidin (an approved drug) obtained from citrus fruits fitted very well into and blocked the substrate binding site, resulting as the top scorers. In particular, hesperidin hits showed up multiple times, suggesting it has many modes of binding. Both hesperidin and diosmin were attributed only mild, occasional and reversible adverse reactions.

Another computational and in vitro and in vivo study found that multiple flavonoids abundant in citrus peels have the potential to cooperate to prevent the SARS-CoV-2 infection and restrain its harmful consequences [65]. In particular, simulated molecular docking showed that naringin, hesperetin

and naringenin, in descending order, have strong binding affinity with the RBD–ACE2 receptor, at a level similar to chloroquine and higher than hesperidin. Moreover, in vitro and in vivo experiments showed the potential of naringin for inhibiting or restraining the expression of the proinflammatory cytokines induced by different disorders through the overreaction of the human immune system, thereby suggesting that naringin could have a potential in preventing cytokine storms associated with severe forms of COVID-19. It appears that integral flavonoids-rich extracts from citrus peels could show simultaneously multiple activities against COVID-19.

Naringin, among all the flavonoids available in the traditional Chinese herb *Exocarpium Citri grandis*, was attributed the greatest potential for application in alleviating the respiratory symptoms caused by COVID-19 [66]. This role was due to several properties, such as antitussive, regulation effect on both mucus and serous components in sputum, improvement of lung function and regulation of pulmonary secretion, and inhibition of the secretion of pulmonary inflammatory factors, thus alleviating acute lung injury. Moreover, naringin was shown to possess therapeutic effects in attenuating pulmonary fibrosis and enhancing the antiviral immune response.

In a further study, a library of phenolic natural compounds (80 flavonoids) was investigated by in silico based screening method against the crystallized form of SARS-CoV-2 main protease ($M^{pro}/3CL^{pro}$) [61]. The importance of $M^{pro}/3CL^{pro}$ derives from its key role in the self-maturation and processing of viral replicase enzymes, thus in virus replication. Hesperidin exhibited the highest binding energy at the active site of SARS-CoV-2, and revealed as the best potential inhibitor of $M^{pro}/3CL^{pro}$ by using a molecular docking approach, closely followed by rutin and diosmin (another citrus flavonoid). Moreover, both hesperidin and diosmin showed a better binding affinity to $M^{pro}/3CL^{pro}$ than nelfinavir, an antiviral widely used in the treatment of HIV, as well as one of the early candidates for the treatment of COVID-19 [67].

Later studies, published in April 2020, provided important confirmation to the potential role of hesperidin against COVID-19. Joshi and coauthors performed an extensive molecular docking study with over 7000 molecules from different classes such as flavonoids, glucosinolates, antitussive, anti-influenza, antiviral, terpenes, terpenoids, alkaloids and other predicated anti-COVID-19 molecules [62]. The three docking targets were $M^{pro}/3CL^{pro}$, involved in virus replication; RNA-dependent RNA polymerase (RdRp), which carries out the synthesis of viral RNA from RNA templates and is involved in the replication and transcription of viral genome; and human ACE2, which is the entry point of the virus. Based on the finding in a previous study [59] that effective molecules should target multiple key proteins, out of all the considered molecules, only 29 were predicted as potentially effective against COVID-19, and among these, several flavonoids showing better binding affinities to the three targets than existing synthetic antiviral drugs. Out of the predicted molecules, hesperidin showed the second highest average binding score across the targets, as well as by far the highest binding score with human ACE2, thus potentially representing one of the most promising molecules for any stage of the infection, as well as the most promising for prevention purposes.

The latter result concerning hesperidin was confirmed in another independent study by Indian authors [68]. They concluded that hesperidin, showing the lowest binding energy with the spike protein fragment of SARS-CoV-2 and its human host ACE2 receptor, can be considered as the most suitable ligand across several phytochemicals typical of Indian medicinal plants.

In another study, plant bioactive compounds were assessed based on their binding affinity with $M^{pro}/3CL^{pro}$ and spike glycoprotein of SARS-CoV-2, by means of a molecular docking approach [63]. The well-known drugs, namely nelfinavir, chloroquine and hydroxychloroquine sulfate, which were widely recommended for clinical trials against COVID-19, were used as a comparison. Hesperidin turned out to have the highest binding score towards both targets, outperforming also the above-mentioned drugs, and significantly, chloroquine and hydroxychloroquine. Other bioactive compounds from citrus fruits, such as rhoifolin, nobiletin, tangeretin, and chalcone, showed good binding affinity.

In a later important development, the Qingfei Paidu Decoction, a formula consisting of 21 components including both herbs and mineral drugs, well known in traditional Chinese medicine, was analyzed in order to explain the mechanisms underlying its observed success in treating COVID-19 patients [69]. It was found that such natural drug contains plenty of bioactive compounds, including hesperidin, neohesperidin, naringin and rutin, and is particularly effective in regulating the innate immune system and preventing cytokine storms through the regulation of the toll-like signaling pathway. Synergistic effects were preliminarily revealed, although requiring further investigation along with the possibility of additional actions against COVID-19.

## 4. Extraction of Hesperidin and Other Citrus Bioactive Compounds

### 4.1. Extraction Methods

Hesperidin is mostly extracted from the citrus peel as a flavonoid complex with 60–70% hesperidin concentration via a time-consuming process, using large amounts of mineral acid and mineral base. In particular, the extraction foresees the treatment of the peel with a NaOH solution at pH 11.5, followed by acidification with mineral acid and heating the acid solution at pH 4.2 at 45 °C overnight [70].

Greener production routes include hydroalcoholic extraction of hesperidin from lime peel and subsequent purification over polymeric adsorption resins to increase the recovery efficiency. This method has been demonstrated both on the laboratory and the semi-industrial scale, even though requiring the addition of 10% dimethyl sulfoxide (DMSO) to the extract in order to improve the solubility of hesperidin [47]. Another state-of-the-art, greener hydro-distillation extraction method applied to orange peels allowed the extraction of total polyphenols in the aqueous phase with the yield of about 17% of the original content [71].

Hydrodynamic cavitation methods invariably involve the creation of a periodic depression in a liquid mixture, either by means of the active circulation of the liquid through a nozzle of suitable geometric shape, or moving mechanical parts, such as rotor–stator arrangements, in a still liquid. Vapor filled nano- and micro-bubbles form whenever the liquid pressure falls below the vapor pressure, and subsequently grow and implode under the external force produced by the recovered bulk liquid pressure. The implosion events release extraordinary intense energy pulses, and eventually, pressure shockwaves, hydraulic jets, extreme transient heating, and chemical dissociation reactions. Past studies and comprehensive reviews are available, which explain the above-mentioned mechanisms in great detail [72–74].

In terms of process yield, HC methods were found to outperform alternative long established and emerging methods, including acoustic cavitation, for most of applications including wastewater remediation, water disinfection, and especially extraction of natural products [72,75], showed compliance with the principles of green extraction [76], and demonstrated straightforward scalability [77]. In particular, fixed-reactor arrangements, such as based on orifice plates or Venturi tubes, are easy to construct and operate, reliable and can be easily optimized towards the desired effect [72]. Venturi-shaped reactors are particularly suitable for applications such as extraction of natural products, due to more diffuse bubble implosion events and avoidance of clogging, in comparison to orifice plates.

The HC-based integral extraction of waste citrus peel in water, without any other additive, was demonstrated directly on the semi-industrial scale using waste orange peels (WOP) [78] and waste lemon peels (WLP) [79,80] from citrus fruits organically grown in Sicily. No other processes allowed to extract in just 10 min around 60% (*w/w*) of the overall polyphenol content in fresh WOP [78].

The processing of 6.38 kg of fresh WOP in 147 L of water allowed the extraction of 36.26 g of hesperidin (0.6 wt%) in the aqueous phase. Along with hesperidin, a significant amount of naringin (16.39 g), other flavonoids (2.95 g), essential oils (mainly *d*-limonene) and water-soluble pectin were extracted. The feasibility of process upscaling, up to as much as 42 kg of fresh WOP in 120 L of water,

was proved, pointing to a concentration of hesperidin in the aqueous extract as high as 0.2% (*w/v*), i.e., 2000 mg/L.

Figure 1 shows the pilot device implementing the above-mentioned HC-based processes with WOP and WLP [78–80], including a closed hydraulic loop (total volume capacity around 230 L), a centrifugal pump (7.5 kW nominal mechanical power, rotation speed 2900 rpm), and a Venturi-shaped cavitation reactor, whose geometry was defined in a previous study [81]. The processes were carried out at atmospheric pressure.

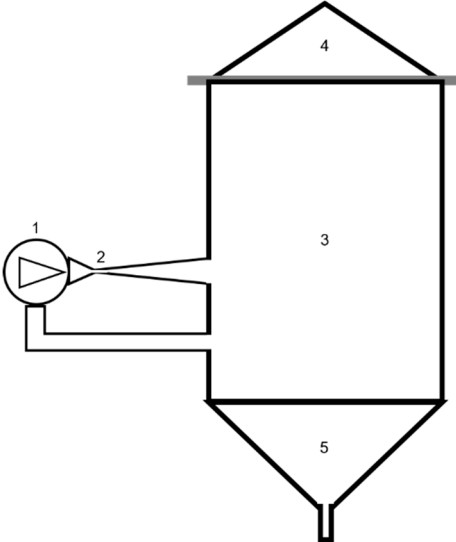

**Figure 1.** Scheme of the experimental hydrodynamic cavitation-based installation. 1—centrifugal pump, 2—hydrodynamic cavitation (HC) reactor, 3—main vessel, 4—cover, 5—discharge.

Such device was used in past studies to carry out innovative beer-brewing [82–85], for which application an industrial-level plant (2000 L) was developed and operated [77,86,87]. Other applications were demonstrated, such as the solvent-free extraction of bioactive compounds from the leaves of silver fir plants [76], and the enhancement of biochar properties [88]. Further applications were designed, such as aimed at the manufacturing of vegetable beverages based on cereals [77], and on oilseeds [87].

Venturi-shaped cavitation reactors were shown to outperform other reactors based on fixed constrictions, such as orifice plates, in the treatment of viscous food liquids [75]. This superiority holds especially with liquids containing solid particles, as well as for the inactivation of spoilage microorganisms [81], and for the creation of oil-in-water stable nanoemulsions [89], all these features being relevant to the processes under study.

*4.2. Properties of the IntegroPectin*

The flavonoids extracted in the aqueous phase by the HC processes, along with essential oils and the water-soluble pectin, were isolated via lyophilization of the aqueous solution, affording a flavonoid-rich pectin dubbed "IntegroPectin" [78–80]. Widely employed in the food industry as the natural hydrocolloid of choice [90], pectin exhibited a broad biological activity, including immunoregulatory, anti-inflammatory, and hypoglycemic activities, for which it has been increasingly used in various pharmacological applications [54]. IntegroPectin extracted from *Citrus sinensis* WOP had a very low degree of esterification (17%), making it particularly appropriate for food, pharmaceutical, and nutraceutical applications [78].

The bioactive properties of the IntegroPectin were thoroughly analyzed for the case of the HC-based processing of WLP. The IntegroPectin showed a very high total polyphenol content that, compared with typical concentration levels in lemon peels, allowed to conclude that most of polyphenols, available

in the raw material and dominated by hesperidin, are concentrated at the surface of the freeze-dried pectin [79].

Accordingly, the IntegroPectin showed exceptional antioxidant properties, based on the oxygen radical absorbance capacity (ORAC) assay as well as on the inhibition of the organic hydroperoxides-induced oxidative stress to human epithelial cells [79]. The IntegroPectin also showed complete lack of cytotoxicity against the same pulmonary epithelial cells up to very high doses (1 mg/mL). These properties were retained even after the application of heat stress, namely 121 °C for 15 min, or 200 °C for 5 min, showing excellent physicochemical stability.

IntegroPectin also showed a strong antibacterial activity against *Staphylococcus aureus*, a Gram positive pathogen which easily contaminates food, with a decrease in the number of viable cells up to two log units [80]. In particular, IntegroPectin outperformed commercial citrus pectin obtained via hydrolysis with mineral acids of dried lemon peel, followed by prolonged separation and purification of the degraded pectic polymer.

It was hypothesized that the antibacterial, antioxidant and the lack of cytotoxicity properties could be attributed to the high concentration of hesperidin and other flavonoids adsorbed and concentrated at the IntegroPectin surface during the freeze-drying process of the aqueous phase, as well as to essential oils, whose presence in the IntegroPectin was confirmed by its intense lemon scent. The antibacterial activity was likely boosted by the micronization of *d*-limonene, first in the form of cavitation-induced nanoemulsion, and then deposited onto the pectin surface. Indeed, it is known that the administration of *d*-limonene in the form of nanoemulsion increases its antibacterial activity by many times [76].

## 5. Discussion and Conclusions

Hesperidin, a flavonoid abundant in citrus peels, was identified as a potentially very interesting molecule in the fight against COVID-19. Its antiviral activity was proven for other viruses, in particular SARS-CoV, thus it could reveal useful also in case of further mutations of SARS-CoV-2.

In the therapeutic use, hesperidin has the advantage of strong binding affinity to all the main viral and cellular targets, outperforming not only other natural molecules, but also antiviral drugs recommended for clinical trials on COVID-19 inpatients. These targets correspond to different stages of the infection, from the entry of the virus into the host human cell, to the transcription of viral genome and virus replication.

The especially great binding affinity with the human ACE2, thus the potential to prevent the virus to spread into the cells, could suggest a special role of hesperidin in prophylaxis. On the other hand, the regular and prolonged administration of hesperidin for prophylaxis would be allowed by its safety, short lifetime in the body, and the absence of cytotoxicity up to high doses.

Other flavonoids, coexistent with hesperidin in citrus peels, also showed good binding affinity to one or more targets, especially hesperetin (the aglycone of hesperidin) and naringin. The latter flavonoid also showed the ability to restrain the proinflammatory overreaction of the immune system, which could help fighting the severe forms of COVID-19.

In this study, we call for the urgent uptake of HC-based processes applied to citrus peels, for the efficient and green industrial production of aqueous extracts and pectin tablets rich in hesperidin. The extraction process takes no longer than 10–15 min, however, including all the necessary steps such as grinding the citrus peels before the inlet to the processing unit, separating the solid residues, and discharging and packaging the aqueous extract, the overall process for a plant with a nominal capacity of about 2000 L could require up to 2 h. Based on the figures exposed in Section 3, undertaking the processing of 500 kg waste citrus peel (as such) in 1500 L water, the process would be able to extract 3 kg of hesperidin per cycle, hence at least 36 kg of hesperidin per day (in 12 cycles). The only additional technological components next to the industrial scale HC-based extractor would be a grinder, a filter/separator, and a lyophilizer, such as those commonly operated at pharmaceutical companies, where they are used to remove solvent from a frozen product by sublimation. After the lyophilization,

IntegroPectin tablets containing the required dose of hesperidin and other flavonoids could be readily produced, due to the low density and open, porous structure of the pectin.

Lyophilization, or freeze-drying, is the main drying technique already adopted both in the pharmaceutical and nutraceutical industries, since it removes the water from sensitive products without damaging them. The operational cost of lyophilization is tightly related to the cost of electricity, which is very low in large countries with important manufacturing bases, such as China, India and Russia. In other countries, pharmaceutical and nutraceutical companies today are virtually all equipped with several hundred kW, or even MW solar photovoltaic rooftops which have cut their electricity bills by 30%–50%.

Under the laboratory conditions of the studies mentioned in this article [78–80], batch drying of IntegroPectin took up to 4 days. However, continuous lyophilization techniques, developed for the coffee industry in the 1960s and reducing drying time and electricity consumption, are in the process of being extended to the pharmaceutical and nutraceutical industries by the booming lyophilization equipment industry [91].

In particular, new continuous freeze-dryers for suspended vials were proposed and demonstrated for pharmaceutical applications, which are cheaper and 6–8 times smaller than conventional batch lyophilizers, based on comparison done on constant throughput [92]. They improve heat transfer uniformity and reduce the primary drying time by 3–4 times, while improving vial-to-vial and intravial homogeneity. The total cycle time was estimated approximately 6 times shorter, because some time-consuming operations, such as filling, are carried out in parallel to the process and do not contribute to increase the cycle time. Such equipment would be particularly well suited for the production of IntegroPectin. However, we acknowledge that further research is recommended, aimed at optimizing and streamlining the industrial-scale IntegroPectin production process, from HC-based extraction and continuous freeze-drying to the manufacturing and packaging of tablets.

Figure 2 shows the main technological components of the proposed process based on the experience gained by authors, although variants are easy to set up, such as replacing the centrifugal pump and the Venturi-shaped reactor with a rotor–stator arrangement, according to specific and local expertise or whatever preference. Moreover, the Venturi-shaped reactor could be realized in accordance with long established rules for circular-section ones [81], or in the form of generally more performing slit Venturi [72], as well as optimized by numerical simulations [93]. Other emerging setups could be used too, such as based on vortex diode [94]. The dosing pump is optional and could be useful for introducing any natural or technical additives. Minor components such as a thermometer and a pressure gauge can be applied to the working vessel and are not shown.

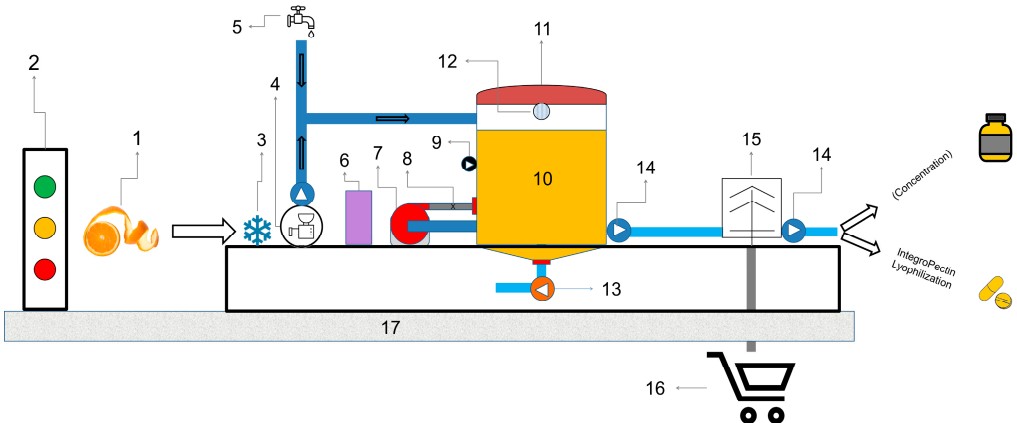

**Figure 2.** Main technological components of the proposed process. 1—citrus waste peel; 2—electronic control panel; 3—ice machine; 4—grinder; 5—water supply; 6—inverter; 7—centrifugal pump; 8—HC reactor, such as a Venturi tube; 9—dosing pump; 10—working vessel; 11—hatchway; 12—washing sphere; 13—lobe pump; 14—multistage pump; 15—filter/separator; 16—discharge of residues; 17—skid.

The product, either in the form of aqueous extracts or pectin tablets, could undergo in vitro, in vivo and clinical trials aimed at assessing the prophylactic or therapeutic activity against COVID-19 and the respective effective doses. As a reference, about one month after the outbreak of COVID-19, Chinese scholars were able to assess the specific antiviral activities of the well-known broad-spectrum antiviral drug remdesivir, and the long-known antimalarial drug chloroquine, against SARS-CoV-2 infecting Vero E6 cells in vitro, including the $IC_{50}$ and the level of the 90% inhibitory concentration ($IC_{90}$) [95]. Remdesivir showed $IC_{50} = 0.77$ μM and $IC_{90} = 1.76$ μM, while chloroquine showed $IC_{50} = 1.13$ μM and $IC_{90} = 6.90$ μM, the latter level clinically achievable as demonstrated in the plasma of rheumatoid arthritis patients who received 500 mg administration. The selectivity index was also reasonably high, about 130 for remdesivir and 88 for chloroquine, suggesting a good safety level.

Since chloroquine is relatively toxic with high doses and prolonged use, as well as its production largely discontinued following the introduction of other antimalarial drugs, about two months later, the same authors proposed hydroxychloroquine as a further potential therapeutic agent against COVID-19 [96]. Hydroxychloroquine was synthesized long ago by introducing a hydroxyl group into chloroquine, and is still widely used in the treatment of inflammatory rheumatic diseases. It is about 40% less toxic than chloroquine, but shows a selectivity index about one third lower and requires higher doses to achieve comparable effectiveness, although still clinically achievable. Finally, hydroxychloroquine shares with chloroquine a good potential to attenuate the inflammatory response, thus potentially offering a broad-spectrum protection from COVID-19.

Despite lower toxicity, one of the main drawbacks of hydroxychloroquine is the well-known side effect of retinopathy, especially in case of prolonged use such as for treating rheumatic disorders and also due to its long half-life and accumulation in tissues and blood, leading to the recent widespread recommendation for lower doses [97,98]. While this could not be such a big issue for therapeutic use against COVID-19, it could jeopardize the use of hydroxychloroquine as a preventive drug. However, the neuroprotective activities attributed to hesperidin, mentioned in Section 2.1 [37,38], might suggest an integrated approach against COVID-19, with hesperidin-rich products and hydroxychloroquine administered together, at respective doses yet to be defined, for both therapy and prevention.

As recalled in Section 2.4, the bioavailability issue after oral administration could impair the performance of hesperidin-rich products during in vivo and clinical trials, which is also the case with research about COVID-19 [63]. Based on the past experience with the HC processing of WOP and WLP [78–80], as well as with other raw materials such as grains [87] and biochar [88], it could be hypothesized that the HC processing relieves the bioavailability issue both for the integral aqueous extracts and the IntegroPectin obtained after freeze drying the extract.

It is likely that the HC processing forms solid fine dispersions, associated with larger available surface and more rapid wetting and dissolution, so that the bioactive compounds can be released efficiently as very fine colloidal particles [46]. If further research will confirm this hypothesis, the integral aqueous extracts could show remarkable bioavailability.

The polyphenols originally contained in the lemon peel were observed to migrate to the surface of the freeze-dried pectin as a result of the HC-based extraction [79]. Thus, it can be hypothesized that the HC processing produces a conjugation of the polyphenols (including hesperidin) onto the pectin macromolecules, resulting in a product similar to that obtained by means of covalent conjugation via a proven preparation method involving epichlorohydrin chemistry [53,54]. In the same way as these conjugates, IntegroPectin is likely to show remarkably higher solubility in water in comparison with neat pectin and polyphenols, along with similar or higher retention of the original biological properties of the conjugated polyphenols. The latter was proven by the exceptional antioxidant activity shown by the IntegroPectin [79].

These topics are recommended for further research, along with the ability of IntegroPectin to regulate the release of hesperidin to the target cells, similarly to the hydrogel matrix obtained after mixing amide pectin with hesperidin and chitosan in a hydroalcoholic solution [52]. In the case of

positive results, IntegroPectin, also in the form of tablets, could be considered for in vivo and clinical trials and, eventually, for mass production.

Finally, since the primary targets of hesperidin and other citrus flavonoids as potential anti-SARS-CoV-2 agents are lung cells, the possibility of formulating such bioactive compounds as an inhalable dry powder (i.e., the IntegroPectin as such) could be considered, similarly to the formulation of naringin by spray-drying with certain amino acids [57]. Further research is recommended on this topic, in particular aimed at investigating whether a substantial fraction of the obtained particles showed aerodynamic diameters <5 μm, thus being able to reach the lung region during inhalation therapy.

**Author Contributions:** Conceptualization, F.M., M.P. and F.Z.; methodology, F.M., R.C. and M.P.; investigation, F.M., M.P. and F.Z.; resources, R.C., F.Z. and M.P.; writing—original draft preparation, F.M. and M.P.; writing—review and editing, F.Z., M.P. and F.M.; visualization, F.M. and F.Z.; supervision, F.M. and M.P. All authors have read and agreed to the published version of the manuscript.

**Funding:** This research received no external funding.

**Acknowledgments:** The authors gratefully acknowledge the National Research Council of Italy (CNR) for invaluable support in these difficult times.

**Conflicts of Interest:** The authors declare no conflict of interest.

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
