# Peer review of "Review of Evidence Available on Hesperidin-Rich Products as Potential Tools against COVID-19 and Hydrodynamic Cavitation-Based Extraction as a Method of Increasing Their Production"

_processes, doi:10.3390/pr8050549_

Round 1

Reviewer 1 Report

The manuscript entitled “Hydrodynamic cavitation-based rapid expansion of hesperidin-rich products from waste citrus peel as a potential tool against COVID-19” is aimed at reviewing the biological properties of the flavanon hesperidin, with particular attention to those potentially able to contrast the COVID-19 and at reviewing the available extraction methods for the compounds.

The paper is well-written and focuses on a very current topic. All efforts to spread knowledge on each possibility to fight the Covid-19 disease are the welcome and well-appreciated.

It is my opinion that the title does not reflects well the content of the paper. In fact, reading the title, I can guess that in the manuscript the authors report a new experiment for extracting hesperidin from citrus byproducts using hydrodynamic cavitation, and then that they tested the extracted molecule against COVID-19 in some way. Actually, the paper is a sort of review of the potentialities of hesperidin with a focus on a specific application for producing a specific product: the IntegroPectin”. The title must better specify that the paper is reviewing these topics.

I have some doubts about the way the authors wrote the “Author Contributions” section. As an example, they state that F.M. performed formal analysis and the same author performed the data curation. But, what analysis and what data? I suggest the authors re-writing this section.

The manuscript is focused on hesperidin. I miss some discussion about its occurrence in nature (for example, is it only present in citrus? Is it only present in peel? What is the reported amount found in the different plant tissue investigated? And so on..).

  • Line 88: the authors state that “Conclusions are set out in Section 5”, but actually they included the Conclusions in section 4, together with the Discussion. Please correct.
  • Line 145: in vitro has to be in italic.
  • Lines 150-152: if available, the authors should report one hypothesis for this different behavior of the other flavonoids with respect to hesperidin.
  • Line 171. It
  • Lines 182-190. This sentences begins “According to recent studies”, but no paper is cited. Even if some references are reported in the following paragraphs, the authors should also add one or, better, more references that report the concepts described in this sentence.
  • Line 226: in vitro and in vivo have to be in italic, here and throughout the manuscript
  • Line 237: in silico has to be in italic, here and throughout the manuscript
  • Line 268-270: this aspect is very interesting, since it looks as the main obstacle to the use of hesperidin for the purpose to contrast the COVID-19. The authors must better investigated on this aspect and enrich the discussion about. For examples, some evidences on bioavailability must be reported, and also some possible ideas and/or proposal to solve it (not only the use of the IntegroPectin, as reported in lines 434-439, but something more. The discussion at lines 425-426 must also be enriched.
  • Line 308: please replace “d-limonene” to “D-limonene”, here and throughout the manuscript.
  • Line 314-348: this part is strongly focused on a specific application for producing a specific product: the IntegroPectin”, differently from the above discussion. It is opinion of this reviewer that this part should be a sub-section of the paragraph 3, with a title clearly related to the specific issue.
  • Line 375: lyophilization is a very expensive drying technique, not always available, and this could be an obstacle to the usability of the system proposed by authors. Have the authors investigated on some possible alternatives to its use, for making their system more easily usable? In this regard, do the authors know how much time is required for a complete lyophilization of the product? Is this required time compatible with the scale-up and consequent higher produced amount proposed? Please discuss this issue.
  • Line 395: or

Reviewer 2 Report

This manuscript is more like a review on how in general the citrus flavonoids are potentially important in pharmaceutical applications, rather than an article paper presenting the hydrodynamic cavitation-based method for extraction of hesperidin-rich products from citrus peel. This manuscript contains very limited information about how the HC-based method is particularly important for the extraction of flavonoids for the treatment of COVID-19. Also, this manuscript has recently been published in Preprints (Meneguzzo, F.; Ciriminna, R.; Zabini, F.; Pagliaro, M. Hydrodynamic Cavitation-based Rapid Expansion of Hesperidin-rich Products from Waste Citrus Peel as a Potential Tool Against COVID-19. Preprints 2020, 2020040152 (doi: 10.20944/preprints202004.0152.v1).). I thus cannot recommend the publication of this manuscript in Processes.

Round 2

Reviewer 1 Report

See attached file

Reviewer 2 Report

The manuscript is much improved after revision and is quite worthy of publication. I appreciate that the authors have changed the title and type of the manuscript. The added information in the text together made the manuscript a comprehensive review.
